# Synergistic Effects of Soil-Based Irrigation and Manure Substitution for Partial Chemical Fertilizer on Potato Productivity and Profitability in Semiarid Northern China

**DOI:** 10.3390/plants13121636

**Published:** 2024-06-13

**Authors:** Lingling Jiang, Rong Jiang, Ping He, Xinpeng Xu, Shaohui Huang, Hanyou Xie, Xiya Wang, Qiying Wu, Xia Zhang, Yi Yang

**Affiliations:** 1Shandong Key Laboratory of Biophysics, Institute of Biophysics, Dezhou University, Dezhou 253023, China; jll@dzu.edu.cn (L.J.); zhangxia@dzu.edu.cn (X.Z.); yangyi@dzu.edu.cn (Y.Y.); 2Institute of Plant Nutrition and Resources, Beijing Academy of Agriculture and Forestry Sciences, Beijing 100097, China; rong_jiang@outlook.com; 3Key Laboratory of Plant Nutrition and Fertilizer, Ministry of Agriculture and Rural Affairs, Institute of Agricultural Resources and Regional Planning, Chinese Academy of Agricultural Sciences (CAAS), Beijing 100081, China; xuxinpeng@caas.cn (X.X.); xiehyys@163.com (H.X.); 18236960892@163.com (X.W.); qiyingwww@163.com (Q.W.); 4Hebei Fertilizer Technology Innovation Centre, Institute of Agricultural Resources and Environment, Hebei Academy of Agriculture and Forestry Sciences, Shijiazhuang 050051, China; shaohui1988@sina.com

**Keywords:** potato, irrigation water, manure substitution, comprehensive evaluation

## Abstract

Soil-based irrigation and the partial substitution of chemical fertilizers with manure are promising practices to improve water and nitrogen (N) use efficiency. We hypothesize that their combination would simultaneously benefit potato production, tuber quality and profitability. A two-year experiment was conducted in semiarid northern China to investigate the combined effects of three water treatments [rainfed (W0), soil-based irrigation (W1), conventional irrigation (W2)] and three N treatments [no N (N0), chemical N (N1), 25% manure substitution (N2)] on these indicators, and to perform a comprehensive evaluation and correlation analysis. The results showed that water and N treatments separately affected all indicators except vitamin C content. Compared to W2, W1 significantly increased water productivity by 12% and irrigation water use efficiency (IWUE) by 30% due to 10% lower evapotranspiration and 21% lower water use. However, W1 and W2 negatively affected crude protein content. Conversely, this was compensated by the combination with N1 and N2. There were slight differences between N1 and N2 for all indicators on average across water treatments, while under W1, N2 significantly increased leaf area index (LAI) and N recovery efficiency (REN) by 18% and 29.4%, respectively, over N1. Also, comprehensive evaluations showed that W1N2 performed best, with the highest tuber yield, profit and acceptable quality. This can be explained by the increase in LAI, IWUE and REN due to the positive correlations with tuber yield and net return. Consequently, soil-based irrigation combined with 25% manure substitution had complementary effects on tuber quality and synergistic effects on potato productivity and profitability.

## 1. Introduction

The potato (*Solanum tuberosum* L.) plays a crucial role as food demand continues to increase, as it is the world’s third most important food crop in terms of human consumption, after rice and wheat [1]. China is the largest potato producer in the world, accounting for 32% (5.8 million hectares) of the world area harvest and 25% (94.4 million tons) of the world production in 2021 [2]. Nearly 40% of China’s potato acreage is distributed in the northern region of the country, greatly contributing to local food diversification, income and employment [3]. However, potato yield in northern China is below the world average due to the lower fertility of sandy soils and the diminishing availability of irrigation water in the semiarid regions.

The severe water shortage in northern China dictates the prioritization of improving irrigation water use efficiency (IWUE) in agricultural production [4]. A drip irrigation system, as one of the most effective methods, tends to improve the IWUE of potatoes grown on sandy soils [5]. However, excessive water use with a drip system is particularly problematic for conventional irrigation by farmers due to inadequate irrigation scheduling [6]. In contrast, irrigation scheduling based on the soil moisture using the drip method (soil-based irrigation) delivers appropriate water quantities at the proper timing to match crop demand and maximize water conservation [7,8]. This structured irrigation timing and amount could keep soil water content in the root zone at field capacity, contributing to water and N retention for plant uptake, thereby reducing leaching and increasing water and N use efficiency [9]. However, there is still a research gap on the effects of soil-based irrigation on water and N use efficiency in potato cropping systems in northern China. 

Potato production in northern China has a 21% higher usage of chemical N fertilization than the recommended rate, implying the lower N use efficiency of potatoes in the region [10]. Manure substitution for partial chemical fertilizers as an efficient practice is widely implemented, and has positive effects on crop productivity and N use efficiency, while alleviating the environmental challenges caused by the overuse of chemical fertilizers in agroecosystems [11,12]. However, its impact on water productivity and IWUE in semiarid regions remains unknown, particularly in an agropastoral ecotone of northern China, where potato production accounts for about 47% of the total food crop production [13]. In addition, producers are willing to implement this practice only if it is beneficial on an economic scale [14]. Therefore, one of the most important aspects is to investigate its economic benefit, based on soil type and climatic conditions, to increase producers’ willingness. 

Previous studies on potato cropping systems mainly focused on the independent effect of soil-based irrigation [15] and partial manure substitution [16,17,18]. Given their independent positive effects on tuber yield, plant growth and resource use efficiency, we hypothesize that their combined effects would simultaneously benefit potato production, tuber quality, water and N use efficiency, and economic return. Moreover, there are few reports on the relationship between water and N use efficiency with potato tuber yield and profitability. Therefore, a two-year experiment was conducted to (1) present the water-saving potentialities of soil-based irrigation, (2) investigate and evaluate the effect of soil-based irrigation combined with partial manure N substitution on LAI, tuber yield and quality, water and N use efficiency, and net return, and (3) to determine the correlations between the above indicators.

## 2. Results and Discussion

### 2.1. Production Performance

#### 2.1.1. Leaf Area Index

The LAI depended on water and N, and their combined treatments, but not on their interaction (Table 1). LAI was the highest (6.6) in potatoes grown with W2N2 treatment, while it was the lowest (3.7) in potatoes grown with W0N0 treatment (Figure 1). The absence of water and N inputs resulted in low LAI [19]. The insignificant interaction was because the LAI response to water treatments was independent of the N treatments. These results indicate that both water and N act separately and additively.

The main effect of water treatments on LAI followed the pattern W2 > W1 > W0 (Figure 1). Compared to W0, W2 significantly increased LAI by 28.2%, while W1 showed no significant increase. This is attributed to the non-significant difference in the first year, while W1 significantly increased LAI by 25.6% compared to W0 in the second year (Appendix A). The reason for this can be explained by the fact that the water drainage under W1 in 2020 (82.5 mm) was higher than that in 2021 (0 mm), and therefore the less N leaching in 2021, the more significant increase in LAI. Similarly, Akkamis et al. [19] reported a significant increase in LAI for N input treatments over non-N treatments in the year with less N leaching. Therefore, the positive effect of soil-based irrigation gradually appeared with decreasing water and N leaching.

The main effect of N treatments showed that N inputs resulted in a higher LAI of potatoes, following N2 > N1 > N0 (Figure 1). N1 and N2 significantly increased LAI by 39.8 and 51.6% compared with N0, which is consistent with the results reported by Zhang et al. [20]. There was no significant difference in LAI between N1 and N2 when averaged over water treatments and years, while N2 significantly increased LAI over N1 by 16.0% under W1 in the second year. This suggests that the 25% manure substitution had a greater positive effect on LAI than 100% chemical fertilization under soil-based irrigation in the year without water and N leaching.

#### 2.1.2. Tuber Yield and Commodity Ratio

Tuber yield and commodity tuber ratio varied significantly with the water and N combined treatments (Figure 2). Tuber yield and commodity tuber ratio were lowest (16.5 t ha^−1^ and 39.5%) under the W0N0 treatment, while they were highest (36.6 t ha^−1^ and 73.3%) under the W1N2 treatment. The significant interactions of water and N on the commodity tuber ratio further demonstrated that the benefit of W1 depended on N2. 

Tuber yield and commodity tuber ratio were influenced by water, N and year, and depended on the interactions of W × Y and N × Y (Table 1). The main effect of water treatments on tuber yield followed the pattern W1 > W2 > W0, with average increases of 77.9% and 74.8% for W1 and W2, respectively, compared with W0. The response pattern observed for commodity tuber ratio was consistent with that for tuber yield. There were no significant differences between W1 and W2, suggesting that conventional irrigation with 21% higher water use failed to increase potato production compared to soil-based irrigation. A previous study also reported that conventional irrigation resulted in 38% more water use compared to soil-based irrigation, leading to a lower or similar yield [7]. Therefore, conventional irrigation, which causes over-irrigation and the wastage of precious freshwater, should be alternated with soil-based irrigation for potato production in semiarid regions.

The main effect of N treatments on tuber yield followed the pattern N2 > N1 > N0, with insignificant differences among them (Figure 2). Nakidakida et al. [17] also found that there was no significant difference in tuber yield between 25% manure substitution and 100% chemical fertilization in two years. However, they reported that N input treatments significantly increased tuber yield compared to non-N treatments [17], which is inconsistent with the results of the current investigation. This was explained by the higher tuber yield with N0 under the W1 and W2 treatments than that with N1 and N2 under the W0 treatments (Appendix A). This might be due to the sufficient N supplied by the soil in two years. Some similar findings were reported in previous studies [21,22,23], which showed that tuber yield showed an insignificant response to the N inputs under irrigated conditions. These results indicated that tuber yield was less sensitive to N input than water input. A similar pattern for tuber yield was recorded for commodity tuber ratio, with a significant increase of 22.2% higher commodity tuber ratio for N2 than N0 (Figure 2). This result was consistent with that obtained by Nakidakida et al. [17], who reported that the 25% manure substitution treatment significantly increased commodity tuber yield compared to the non-N treatment. Moreover, the significant interactions of N and year indicated that N2 had a greater positive effect in the second year (Appendix A). As potato tubers are used for fresh consumption in this region, the high commercial tuber ratio with 25% manure substitution is more advantageous than 100% chemical fertilization, with an increasing trend over the years.

Comparing the two years, tuber yield was higher in the second year (Appendix A), mainly due to the greater positive effect of W1 and N2 in the second year (Appendix A). This result indicated that the positive effect of water and N inputs on tuber yield increased with the experimental years. This is consistent with the previous findings that the effect of partial manure substitution on crop yields tended to increase with the duration of the experiments [12]. However, the result of the comparison of two years of commodity tuber ratio was opposite to tuber yield. This is due to the greater reduction in commodity tuber ratio in the W0 and N0 treatments in the second year (Appendix A). The main explanation is that persistent soil water and N depletion limit the production of commodity tubers [21]. These results indicated that the negative effect of no water and N inputs was observed in the short term (two years), whereas the positive effect of water and N inputs needs a long-term observation.

### 2.2. Tuber Quality Performance

The result showed that the water and N treatments had significant effects on crude protein content, but not on vitamin C (VC) content (Table 1). This may be due to the fact that the VC content of tubers is genetically controlled. Both VC and crude protein content were varied with the year, with higher levels in the second year (Appendix A). A 11.9% lower VC value in 2020 than in 2021 is mainly due to the fact that the tubers in 2020 were measured after one month in cold storage (at 4 °C). This is because the tubers’ VC content dropped rapidly with longer intervals of cold storage [24]. The 7.5% higher crude protein content in 2021 than in 2020 was related to the 9% increase in plant N uptake (Appendix A).

The main effect of water treatments on crude protein content followed the pattern W0 > W2 > W1, with significant decreases of 11.1% and 14.6% for W2 and W1, respectively, compared with W0 (Figure 3). Similar findings were reported in previous studies, where irrigation water had a negative effect on tuber crude protein content [25]. This may be related to the fact that the increased irrigation amount improved the flow of water from xylem to tuber, resulting in a dilution of the crude protein content [26]. The main effect of N treatments followed the pattern N2 > N1 > N0, with significant increases of 17.2% and 16.7% for N2 and N1, respectively, compared with N0, indicating that N inputs had a positive effect on tuber crude protein content. Especially under N2 conditions, all three water treatments were in the same statistical group with 2.1–2.2% crude protein content, slightly lower than the highest value (2.3%). Therefore, the 25% manure substitution had a compensatory effect, minimizing the negative impact of water irrigation on crude protein content.

### 2.3. Water Productivity and Irrigation Water Use Efficiency

WP and IWUE were significantly affected by the combined treatments (Figure 4). The highest WP and IWUE were recorded with the W0N2 and W1N2 treatments, respectively. This result indicates that N2 has the potential to improve water productivity under rainfed conditions and increase irrigation water use efficiency under soil-based irrigation.

Water, N, and year, individually, had significant effects on the WP and IWUE of the potatoes, with insignificant interaction effects between the three factors (Table 1). The main effect of water treatments on WP demonstrated the following: W0 > W1 > W2, with reductions of 16.1% and 10.9% for W2 compared to W0 and W1 (Figure 4). W2, with the highest water use, resulted in the highest evapotranspiration (ET) and consequently the lowest WP and IWUE (Appendix A). Compared with W2, W1 significantly improved WP and IWUE by 12.2% and 29.6%, respectively, due to a 10% reduction in ET and a 21% reduction in water use (Figure 4 and Table 2). The current results are consistent with the findings of previous studies [15,27,28], which may be attributed to stomata closure under low water use, resulting in a decrease in ET and consequently an increase in WP [23]. Therefore, soil-based irrigation should be promoted as an alternative to conventional irrigation for potato production to efficiently use water and cope with the increasing water scarcity in semiarid northern China.

The main effect of N treatments on WP followed the pattern N2 > N1 > N0 (Figure 4), indicating that WP was improved by N inputs, consistent with previous findings [23,29]. N2 and N1 significantly increased WP over N0 by 16.1% and 10.9%, respectively, but there was no significant difference between N1 and N2. Contrary to our research, Zhai et al. [30] stated that 15% and 30% organic fertilizer substitution significantly improved WP by 10.4% and 11.1% compared to chemical fertilization alone. The reason for the insignificant increase with N2 in this study is the slight increase in the first year, while in the second year there was a significant increase of 14.5% compared to N1 (Appendix A). The response pattern of IWUE to N treatments was similar to that of WP, but there was no significant difference between N treatments. This is attributed to the fact that the IWUE of N0 under W1 was higher than that of N1 and N2 under W2 (Figure 5S). Accordingly, the N input had a positive effect on WP, and IWUE was more sensitive to water input than to N input.

Comparing the two years, WP and IWUE were higher in the second year (Appendix A). This appears to be mainly due to the higher tuber yield (Appendix A) and lower irrigation water in 2021 (Table 2). The lower irrigation water for W1 was due to the deeper location of the soil moisture sensor (Appendix A), while for W2 this was due to the limited water available for irrigation in the area. Therefore, in the context of increasing water scarcity, soil-based irrigation with the proper placement of soil moisture sensors was beneficial to water conservation.

### 2.4. Nitrogen Use Efficiency

The IEN and REN of potatoes depended on water and N as well as their combined treatments, and IEN also depended on the years and interactions of W × N, W × Y, and N × Y (Table 1). The highest IEN (246 kg kg^−1^) and REN (38.2%) were observed in the W1N0 and W1N2 treatments, respectively (Figure 5). Although the IEN for the water and N input treatments (177–187 kg kg^−1^) were lower than that of W1N0, they were higher than the highest value (110 kg kg^−1^) reported by Nakidakida et al. [17]. The significant interaction effect between water and N on IEN suggests that N management should consider water management, as N use efficiency is closely related.

The main effect of water treatments on IEN and REN followed the pattern W1 > W2 > W0. W1 and W2 significantly increased IEN by 29.7% and 25.8% and REN by 16.5 and 13.9 percentage points, respectively, compared to W0. These results were in line with previous studies [31], indicating that water input had a positive effect on IEN and REN. This positive effect of W1 and W2 on IEN was greater in the second year (Appendix A). However, W1 showed the insignificant increases in both IEN and REN from W2. This was mainly because the plant N uptake and N uptake gap (total N uptake by plants in N-applied plots minus total N uptake by plants non-N-applied plots) were statistically equivalent between W1 and W2 (Appendix A).

The main effect of N treatments on the IEN of the potatoes followed the pattern N0 > N1 > N2, with a significant decrease of 18.2% and 19.6% for N1 and N2, respectively, compared with N0. The lower IEN suggested that the plant N uptake was not efficiently converted into tuber yield [17]. These results indicated a negative effect of N1 and N2 on IEN, which was attributed to the slight increase in tuber yield and the marked increase in plant N uptake (Figure 2 and Appendix A). However, these negative effects were mitigated in the second year due to the higher tuber yield in 2021 (Appendix A), reflecting the need for further long-term observation. Considering the acceptable IEN for both water and N input treatments, the negative effect of the N input would be compensated to some extent by the water input.

Regarding the response of the N treatments on REN, the main effect followed the pattern N2 > N1. The higher REN under N2 may be related to its elevated plant N uptake, resulting in a higher N uptake gap, which was in agreement with the results from other studies [10,32]. However, there was no significant increase in REN for N2 from N1 because of the lower REN of N2 under W0 (Appendix A). Nevertheless, under the W1 treatment, N2 significantly increased REN over N1 by 6.6 and 10.8 percentage points in 2020 and 2021, respectively. These results suggest that 25% manure substitution is beneficial to improve REN under soil-based irrigation. Therefore, soil-based irrigation combined with 25% manure substitution was beneficial for efficient N use with the highest REN and an acceptable IEN, and the combined effect would be better along with increasing experimental years.

### 2.5. Net Return

The net return depended on water, N, and year, but not on their interactions (Table 1). The main effect of water treatments followed the pattern W1 > W2 > W0, with significant increases of 272.7% and 245.7% for W1 and W2, respectively, compared with W0 (Figure 6). The main effect of N treatments followed the pattern: N2 > N1 > N0, with increases of 39.6% and 58.9% for N1 and N2, respectively, compared with N0. These results indicated that the water input had a greater effect than the N input. Comparing the two years, the net return was 32.2% higher in 2021 than in 2020, which was due to the 30.6% higher income from tuber yield and the 33.5% lower irrigation cost (Appendix A).

However, there was no significant increase in the net return of W1 compared to W2, mainly because the net return of W1 under N0 was lower than that of W2 under N1 and N2 (Appendix A). In contrast, the insignificant increase in the net return for N2 over N1 was due to the lower net return for N2 than for N1 in the first year (Appendix A). In the second year, N2 significantly increased the net return compared to N1 by 4142 and 7807 CNY ha^−1^ under W0 and W1 treatments, respectively. This is consistent with the results reported by Zhai et al. [30], who indicated that 15% and 30% organic fertilizer substitution improved net benefits by 2257 and 1404 CNY ha^−1^, respectively, compared to chemical fertilization alone. Considering the significant variation of net return among treatments, the highest value, 24,818 CNY ha^−1^, was observed in the W1N2 treatment (Figure 6). Therefore, the combined effect of soil-based irrigation and 25% manure substitution promotes the maximization of economic return.

### 2.6. Comprehensive Evaluation for Different Water–Nitrogen Combined Treatments

The results showed that the W1N2 treatment ranked first under both the TOPSIS and PCA methods, with comprehensive evaluation values of 0.78 and 0.75 (Table 3). This result strongly indicated that soil-based irrigation combined with 25% manure substitution was optimal to balance the objectives of productivity, quality, efficiency and profitability. The adoption of the TOPSIS and PCA methods in this study produced inconsistent ranks for W2N1 and W2N2 (Table 3). A previous study [33] also reported similar results of inconsistency using the TOPSIS and PCA methods, due to the different calculation principles of the two methods. Nevertheless, a positive correlation was identified between the comprehensive effect values of TOPSIS and PCA (correlation coefficients over the two years were 0.9426, *p* < 0.001), confirming the reliability of the evaluation results. Notably, the REN under the optimal treatment (W1N2) in this study was reported to be 38.2%, an increase of 7.3 percentage points compared to the farmers’ practice in northern China (W2N1), and double the REN for the farmers’ practice in Japan (18.4%) [17]. The improvement in REN suggests a reduction in N losses through nitrate leaching and nitrous oxide and ammonia emissions, thereby contributing to the mitigation of groundwater pollution and climate warming, and other environmental challenges [34,35]. Given the potential environmental benefits, we will further estimate the reduction in reactive N losses and greenhouse gas emissions under the combined treatment of soil-based irrigation and 25% manure substitution.

### 2.7. Correlation Analysis between Potato Productivity, Water and Nitrogen Use Efficiency, and Profitability

The correlation analysis results are shown in Figure 7. There were significant positive relationships between LAI and tuber yield, REN and net return, which can be used to explain that the highest tuber yield, REN and net return under the W1N2 treatment, were due to the LAI being the greatest. A previous study reported similar results, showing a strong correlation between yield and LAI [36]. Therefore, LAI was the primary reason for the increase in tuber yield, N recovery efficiency and profit, suggesting LAI can be used as one of the important indicators to guide potato breeding for high yield, high efficiency and high profit.

Additionally, both tuber yield and net income had significant positive correlations with commodity tuber ratio, IWUE, IEN and REN. Thus, the improvement of irrigation water and N use efficiency could contribute to increasing potato production and profitability. These findings would help farmers directly understand the effects of water–N management in terms of tuber yield and economic return (two relatively easy-to-understand concepts), which in turn would encourage farmers to actively implement the practices of soil-based irrigation combined with 25% manure substitution.

## 3. Materials and Methods

### 3.1. Field Experimental Site

The field experiment was conducted during the growing seasons of 2020 and 2021 at Wuchuan Dryland Agricultural Experimental Station (41°08′ N, 111°17′ E, alt. 1570 m), in Hohhot City in the northern China. The region’s climate is classified as a semiarid temperate continental monsoon, provisioned by a duration of 3092 h of annual sunshine and a wide diurnal temperature range, suitable for potato growth. During the growing season (from May to September) in the past 40 years, the mean air temperature was 16.5 °C and precipitation was 296 mm. The daily air temperature and precipitation over two years are shown in Figure 8. The 0.2 m layer of soil texture is a well-drained sandy loam, consisting of 57% sand, 25% silt and 18% clay. The soil pH, bulk density, organic matter, total N, available P, and available K were 8.5, 1.34 g cm^−3^, 2.3%, 1.4%, 14.6 mg kg^−1^, and 85.8 mg kg^−1^, respectively.

### 3.2. Field Experimental Design and Field Management

The experiment was a 3 × 3 factorial strip-plot design with water irrigation regimes as the row factor and N fertilization strategies as the column factor, totaling nine treatments with three replicates. The water irrigation regimes included no water irrigation (W0), a soil-based water irrigation regime (W1), and conventional water irrigation pertaining to the intuition of the farmers (W2). The total irrigation amount and irrigation scheduling per season for each irrigation regimes are presented in Table 2 and Appendix A. The N fertilization strategies included no N fertilization (N0), chemical N fertilization at the rate of 210 kg N ha^−1^ as recommended by the Nutrition Expert Tool for the potato crop (N1) [10], and a substitution of 25% chemical N with sheep manure at the same total N rate as N1 (N2). The same amounts of phosphate and potash were applied to all treatments. The fertilizer application rates for all treatments are shown in Table 2. 

Rotary tillage was carried out one month before planting to improve the soil environment. Tuber pieces weighing ~50 g each, cut from the potato seeds, variety “Huasong NO.7” (Inner Mongolia Huasong Potato Industry Co. Ltd., Ulanqab, China), were planted at a depth of 15 cm with a density of 50,000 seeds per hectare on 4 May 2020 and on 1 May 2021. After emergence, 3–5 cm of soil was layered on top of the beds for greater tuber formation and bulking. The tubers were hand-harvested after planting, on 16 September 2020 and 12 September 2021, respectively. During both growing seasons, other than irrigation and fertilization, the agronomic practices including the prevention and control of weeds, and plant diseases and insect pests were consistent for all treatments.

### 3.3. Irrigation and Fertilization Schedule

Following planting, drip tapes (inner diameter 16 mm, emitter distance 0.2 m with a flow rate of 4 L h^−1^) were placed on the beds and irrigation treatments began on day 44 after planting in 2020 and day 40 after planting in 2021. Appendix A shows the irrigation amount and timing of W1 and W2, such that W2 displays a pattern of consistency. The irrigation amount for W1 at each time (I_e_) was estimated using the following equation [37]:(1)$$I_{e}=10\times \rho b\times \mathrm{SD}\times \left({\mathrm{SWC}}_{\mathrm{upper}}-{\mathrm{SWC}}_{\mathrm{lower}}\right)\times P/\eta$$
where ρb denotes soil bulk density; SD denotes wetted soil depth; SWCupper and SWClower, respectively, denote the upper and lower limit of soil water content (SWC) set based on the previous study [38,39] as shown in Table 4; P denotes soil wetting proportion with a value of 0.7; η denotes the utilization coefficient of the drip irrigation system with a value of 0.9. The irrigation amount for each plot was managed by a flow-meter and a gate valve (YS-20, Jiangsu Yongsheng Instrument Ltd., Suzhou, China). Irrigation timing for W1 was triggered at the lower limit of the average SWC within the root zone, measured hourly by three SWC sensors (S-SMC-M005, Onset Computer Corporation, Bourne, MA, USA). The sensors were positioned 15 cm from the center of the bed, at a depth of 15 cm in 2020 and a depth of 20 cm in 2021 (Appendix A). The strategic 5 cm increase of the SWC sensor depth in 2021 from 2020 was suggested by Yang et al. [40] and Ahuja et al. [28], to represent the spatial variability of SWC within potato fields. To induce tuber maturity and prevent secondary growth, irrigation was suspended two weeks before harvest.

The chemical fertilizer used consisted of urea (46% N) of N, triple superphosphate (46% P_2_O_5_) of P, and potassium chloride (60% K_2_O) of K. The manure fertilizer used in N2 treatments was a fully decomposed sheep dung sourced locally, consisting of 45% water content, 1.64% N, 0.31% P_2_O_5_, and 1.93% K_2_O. Strip fertilization was applied at the planting and tuber formation stages, and a fertilization tank was added within the irrigated plot at the tuber bulking stage. The percentages of the three fertilization timings of N were 30% at planting, 30% at tuber formation, and 40% at tuber bulking. The P fertilizer and manure were applied at planting. With regard to K fertilization, the percentage in 2020 was similar to that of N fertilization in 2020. In 2021, 50% of K fertilizer was applied at planting stage, and the remaining 50% at tuber formation due to the poor solubility of potassium chloride. 

### 3.4. Sample Collection and Analysis 

#### 3.4.1. Leaf Area Index

Three representative plants were sampled in the middle bed of each plot during the tuber bulking stage. The leaf area was determined by the punch weighting method. Leaf area index was calculated according to Wang et al. [41].

#### 3.4.2. Tuber Yield and Commodity Ratio

Plants were harvested from the two center-beds of each plot (Figure 8). The tubers were weighed to determine total yield, then sorted to identify commodity tubers weighing over 150 g and free from diseases and pests. The commodity ratio is the percentage weight of commodity tubers in the total tuber yield. 

#### 3.4.3. Water Productivity and Water Use Efficiency

The evapotranspiration (ET, mm), water productivity (WP, kg m^−3^) and irrigation water use efficiency (IWUE, kg m^−3^) were calculated using the following equations [42]:(2)SWS=SWC×ρb×SD
(3)ET=∆SWS+P+I−R−D
(4)$$\mathrm{WP}=\frac{Y}{\mathrm{ET}\times 10}$$
(5)$$\mathrm{IWUE}=\frac{Y}{I\times 10}$$
where SWS denotes soil water storage (mm); SWC denotes the soil water content (%) at 0–60 cm soil depth (SD); ρb denotes soil bulk density; ∆SWS (mm) denotes the change of SWS before planting and after harvesting; Y denotes the tuber yield (kg ha^−1^); P denotes precipitation, I denotes irrigation, R denotes runoff (negligible due to the flatness of the experimental field); and D denotes drainage amounts (mm) measured using an in situ monitoring device in each plot (Figure 9). In 2020, the average water drainage of W0, W1, and W2, respectively, was 0 mm, 82.5 mm, and 89.9 mm. In 2021, the average water drainage of all treatments was 0 mm.

#### 3.4.4. N Use Efficiency

Three successive plants were selected from the harvest as samples to measure the N uptake in each plot. The samples were divided into stem, leaf, root and tuber, and oven-dried at 75 °C until their weights became constant. They were then weighed to determine the dry matter. The dried samples were ground and filtered through a 1.0 mm sieve to measure the N content using the combustion method with an elemental analyzer (Vario Macro CNS, Elementar, Germany). The N uptake was calculated from the dry matter and N content. The internal use efficiency of N (IEN) and the recovery efficiency of N (REN) were calculated using the following equation [17]:(6)$$\mathrm{IEN}=\frac{Y}{U_{N}}$$
(7)$$\mathrm{REN}=\frac{{∆U}_{N}}{N_{\mathrm{fert}}}$$
where Y denotes the tuber yield (kg ha^−1^); U_N_ denotes the N uptake by aboveground biomass (kg N ha^−1^); and ∆U_N_ denotes the increase in U_N_ due to an increment of applied fertilizer N (N_fert_, kg N ha^−1^). A higher IEN indicates the greater utilization of N by the plant for growth and yield. A higher REN indicates the greater capacity of the plant when it comes to ingesting N from the soil, while also implying less N pollution from an environmental perspective.

#### 3.4.5. Tuber Qualities

Three fresh tuber samples were selected to measure vitamin C content using the 2,6-dichlorophenol titration method. The crude protein content was calculated by multiplying the tuber N content by the conversion coefficient of 6.25.

### 3.5. Economic Benefit Analyses

The net return is the difference between the income from the tuber yield and the cost of goods sold (including fertilizers, seeds, pesticides and irrigation water), calculated using the following equation:(8)$$\mathrm{Net}\;\mathrm{return}=\left(Y_{1}\times P_{1}+Y_{2}\times P_{2}\right)\;-\left(F_{i}\times P_{i}+S\times P_{\mathrm{seed}}+I\times P_{\mathrm{irri}}+C_{\mathrm{pest}}\right)$$
where Y_1_ and Y_2_ denote commodity and non-commodity tuber yields (kg ha^−1^); P_1_ and P_2_ denote the selling prices of commodity tuber and non-commodity tuber (1.1 and 0.3 CNY kg^−1^ in 2020, 1.3 and 0.35 CNY kg^−1^ in 2021); F_i_ and P_i_ denote fertilization rates and fertilization cost (the cost of manure, urea, tripe superphosphate and potassium chloride were 120, 2, 3 and 3 CNY kg^−1^ in 2020, and 120, 3, 3.5 and 4 CNY kg^−1^ in 2021, respectively); S and P_seed_ denote seed rate (2000 kg ha^−1^) and seed cost (3 CNY kg^−1^ in 2020 and 3.5 CNY kg^−1^ in 2021); I and P_irri_ denote irrigation amount and irrigation cost (2.5 CNY mm^−1^); and C_pest_ is the cost of pesticide (0 and 900 CNY ha^−1^ in 2020 and 2021, respectively).

### 3.6. Comprehensive Evaluation System and Methods

The LAI, tuber yield, commodity tuber ratio, crude protein content, WP, IEN and net return were considered to evaluate the comprehensive performance of each treatment in the objectives of productivity, quality, efficiency and profitability (Figure 10). These indicators were selected because they were significantly affected by both water and N and had no missing value for any treatment (Table 1). The comprehensive evaluation was performed using the technique for order preference by similarity to ideal solution (TOPSIS) and principal component analysis (PCA) methods, and the detailed analysis processes were described by Wang et al. [41].

### 3.7. Statistical Analysis

Analysis of variance (ANOVA) and least significant difference (LSD) test were performed using “aov” and “LSD.test” in the R (version 4.1.2 for Windows, R Core Team, Vienna, Austria) to examine the effects of water and N management on the variables. Correlation analysis was performed using the “cor” function to determine the relationship between the variables. The figures were produced using “ggsignif”, “ggplot2” and “corrplot” packages.

## 4. Conclusions

Water and N treatments individually affected potato tuber yield, commodity tuber ratio, crude protein content, water and N use efficiency, and the net returns in this study. Their interactions were not significant for most indicators, due to the non-gradient variation of irrigation water and N application in the water and N treatments. Most indicators in the water and N input treatments performed better in the second year due to the absence of water and N leaching in the second year. The responses of tuber yield and net return to water and N treatments were consistent, implying the combined treatments of water and N had synergistical effects on them. However, inconsistent responses to the water and N treatments were observed for water productivity and the internal efficiency of N and crude protein content, implying the combined treatments of water and N had moderating compensatory effects on them. Especially under combinations of soil-based irrigation and 25% manure substitution, comprehensive evaluations confirmed that it was the optimal alternative to enhance potato productivity and profitability, while balancing tuber quality and water and N use efficiency. Moreover, the correlations between tuber yield and net return with irrigation water and N use efficiency allow producers to understand the combined effects of water and N management from two accessible indicators of tuber yield and profit. These results would increase the willingness of producers to implement soil-based irrigation and 25% manure substitution for potato production.

## Figures and Tables

**Figure 1 plants-13-01636-f001:**
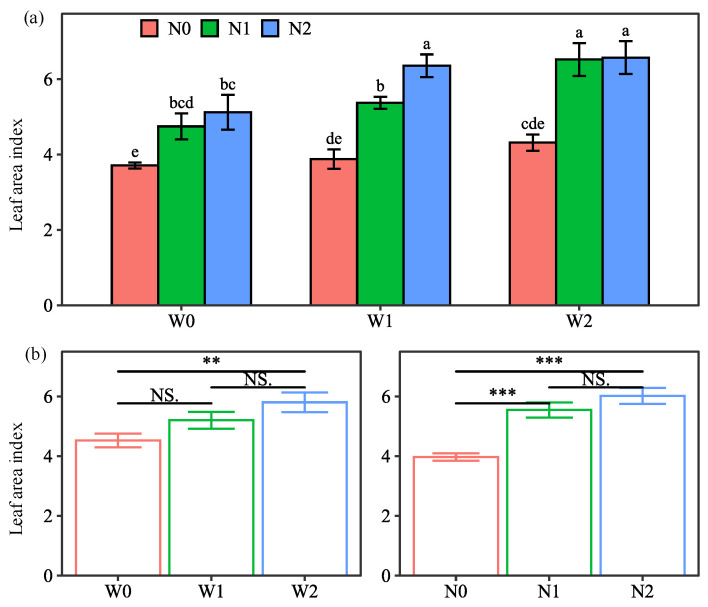
Interaction effect (**a**) and main effect (**b**) of water and N treatments on the leaf area index of potatoes (average of the two seasons). Water treatments: no irrigation (W0), soil-based irrigation (W1), and farmer’s conventional irrigation (W2). Nitrogen treatments: no N application (N0), 100% chemical fertilizer N at 210 kg N ha^−1^ application rate (N1), and 25% substitution of chemical N with manure N (N2). Error bars represent standard errors. Lowercase letters above the error bars represent significant differences between treatments at the *p* < 0.05 level by LSD test. Significant symbols of *** and ** mean significant differences at the level of *p* < 0.001 and *p* < 0.01, respectively, NS means no significant differences.

**Figure 2 plants-13-01636-f002:**
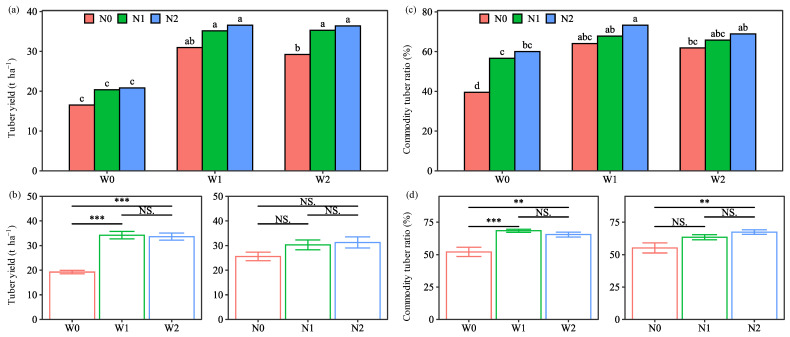
Interaction effect and main effect of water and nitrogen (N) treatments on the tuber yield (**a**,**b**) and commodity tuber ratio (**c**,**d**) of potatoes (average of the two seasons). Water treatments: no irrigation (W0), soil-based irrigation (W1), and farmer’s conventional irrigation (W2). N treatments: no N application (N0), 100% chemical fertilizer N at 210 kg N ha^−1^ application rate (N1), and 25% substitution of chemical N with manure N (N2). Error bars represent standard errors. Lowercase letters above the error bars represent significant differences between treatments at the *p* < 0.05 level by LSD test. Significant symbols of *** and ** mean significant differences at the level of *p* < 0.001 and *p* < 0.01, respectively, NS means no significant differences.

**Figure 3 plants-13-01636-f003:**
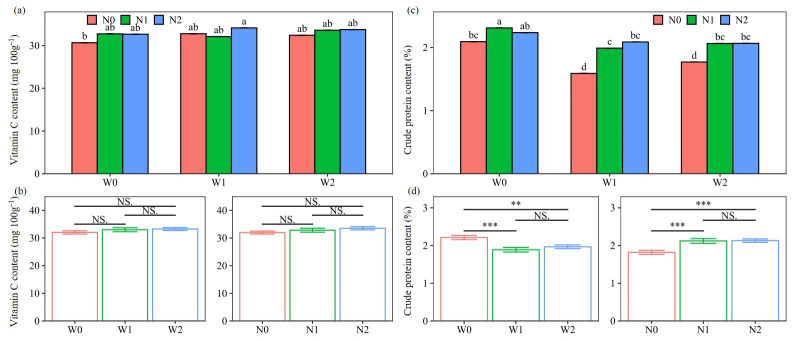
Interaction effect and main effect of water and nitrogen (N) treatments on the Vitamin C (**a**,**b**) and crude protein (**c**,**d**) content of tubers (average of the two seasons). Water treatments: no irrigation (W0), soil-based irrigation (W1), and farmer’s conventional irrigation (W2). N treatments: no N application (N0), 100% chemical fertilizer N at 210 kg N ha^−1^ application rate (N1), and 25% substitution of chemical N with manure N (N2). Error bars represent standard errors. Lowercase letters above the error bars represent significant differences between treatments at the *p* < 0.05 level by LSD test. Significant symbols of *** and ** mean significant differences at the level of *p* < 0.001 and *p* < 0.01, respectively, NS means no significant differences.

**Figure 4 plants-13-01636-f004:**
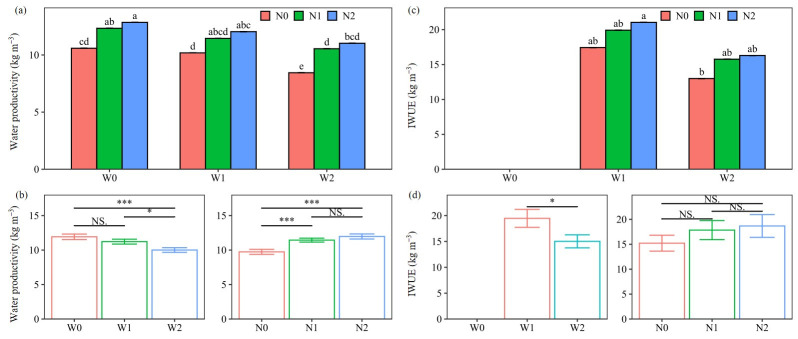
Interaction effect (**a**,**c**) and main effect (**b**,**d**) of water and nitrogen (N) treatments on water productivity and irrigation water use efficiency (IWUE) of potatoes (average of the two seasons). Water treatments: no irrigation (W0), soil-based irrigation (W1), and farmer’s conventional irrigation (W2). N treatments: no N application (N0), 100% chemical fertilizer N at 210 kg N ha^−1^ application rate (N1), and 25% substitution of chemical N with manure N (N2). Error bars represent standard errors. Lowercase letters above the error bars represent significant differences between treatments at the *p* < 0.05 level by LSD test. Significant symbols of *** and * mean significant differences at the level of *p* < 0.001 and *p* < 0.01, respectively, NS means no significant differences.

**Figure 5 plants-13-01636-f005:**
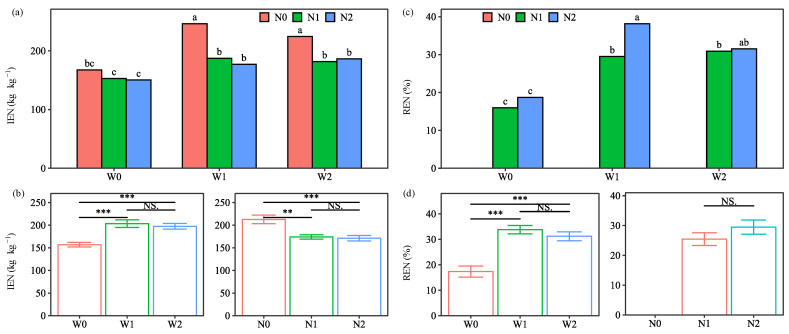
Interaction effect (**a**,**c**) and main effect (**b**,**d**) of water and nitrogen (N) treatments on internal efficiency of N (IEN) and recovery efficiency of N (REN) of potatoes (average of the two seasons). Water treatments: no irrigation (W0), soil-based irrigation (W1), and farmer’s conventional irrigation (W2). N treatments: no N application (N0), 100% chemical fertilizer N at 210 kg N ha^−1^ application rate (N1), and 25% substitution of chemical N with manure N (N2). Error bars represent standard errors. Lowercase letters above the error bars represent significant differences between treatments at the *p* < 0.05 level by LSD test. Significant symbols of *** and ** mean significant differences at the level of *p* < 0.001 and *p* < 0.01, respectively, NS means no significant differences.

**Figure 6 plants-13-01636-f006:**
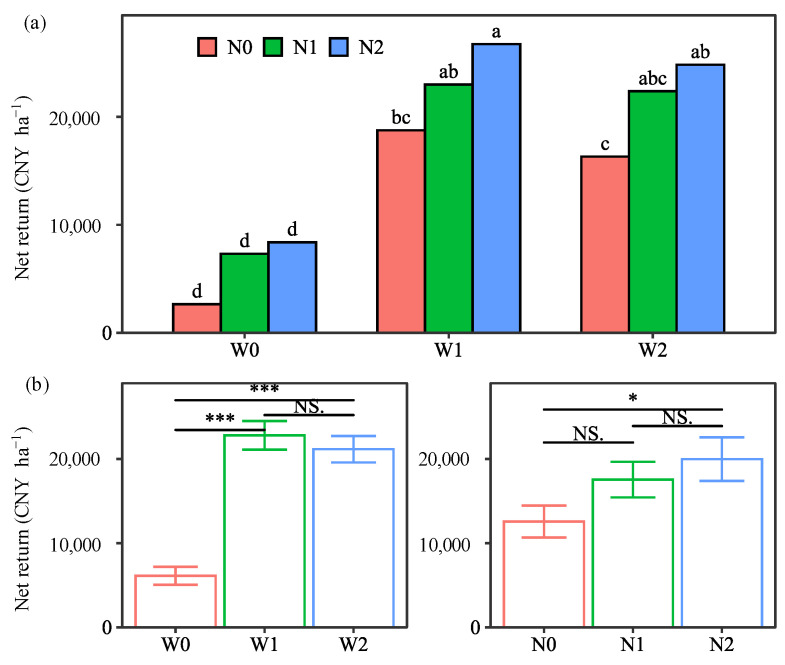
Interaction effect (**a**) and main effect (**b**) of water and nitrogen (N) treatments on the net return (average of the two seasons). Water treatments: no irrigation (W0), soil-based irrigation (W1), and farmer’s conventional irrigation (W2). N treatments: no N application (N0), 100% chemical fertilizer N at 210 kg N ha^−1^ application rate (N1), and 25% substitution of chemical N with manure N (N2). Error bars represent standard errors. Lowercase letters above the error bars represent significant differences between treatments at the *p* < 0.05 level by LSD test. Significant symbols of *** and * mean significant differences at the level of *p* < 0.001 and *p* < 0.05, respectively, NS means no significant differences.

**Figure 7 plants-13-01636-f007:**
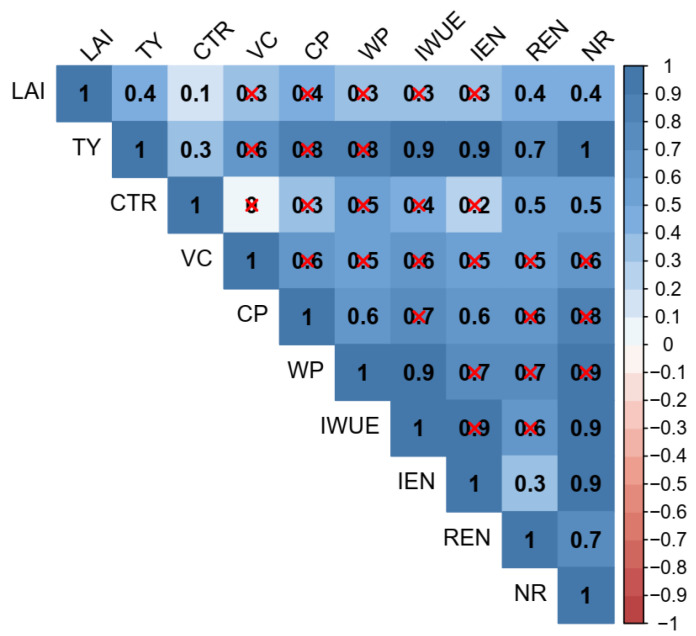
Correlation analysis of leaf area index (LAI), tuber yield (TY), commodity tuber ratio (CTR), vitamin C content (VC), crude protein content (CP), water productivity (WP), irrigation water use efficiency (IWUE), internal efficiency of nitrogen (IEN), recovery efficiency of nitrogen (REN) and net return (NR) at a significant level of *p* < 0.001. The red “×” marks indicate non-significant correlations.

**Figure 8 plants-13-01636-f008:**
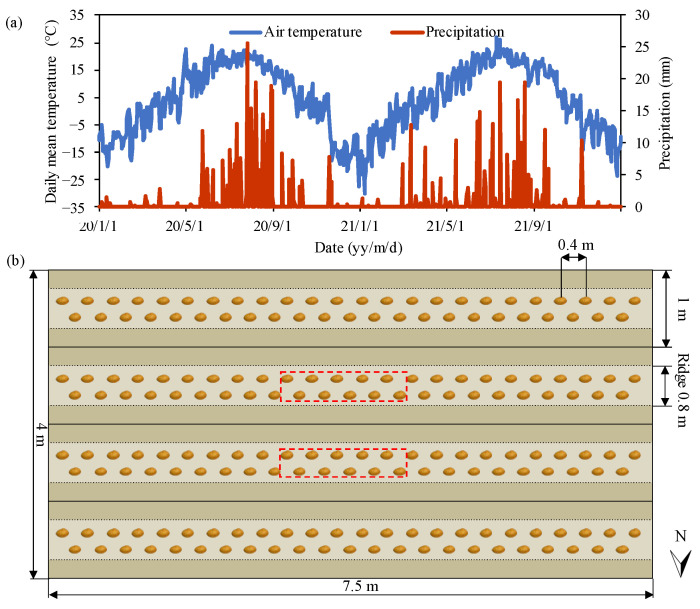
Meteorological data for 2020 and 2021 (**a**) and layout of one of the plots (**b**). The size of each plot was 30 m^2^ (7.5 m × 4 m) included four east–west beds. Potato seed pieces were planted in double rows on each bed. Plants within the red dashed line were harvested to determine tuber yield and quality.

**Figure 9 plants-13-01636-f009:**
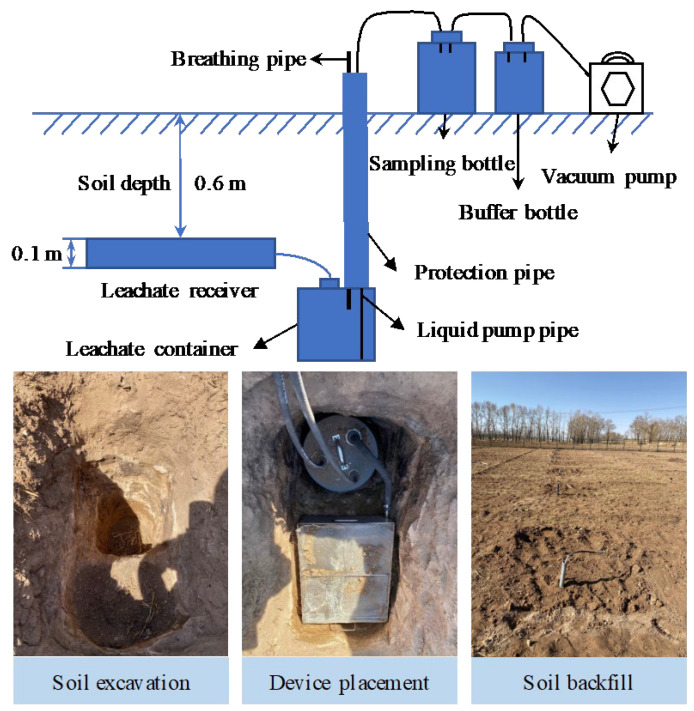
In situ monitoring device for collecting water drainage.

**Figure 10 plants-13-01636-f010:**
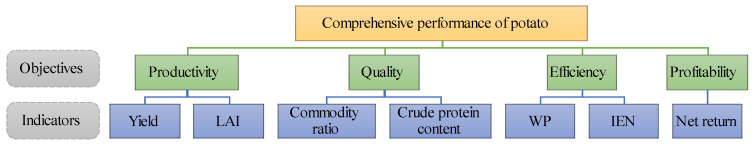
Comprehensive evaluation system of the potatoes. LAI, WP and IEN represent the leaf area index, water productivity and internal efficiency of nitrogen.

**Table 1 plants-13-01636-t001:** Significance levels of the effects of water, nitrogen (N) and year as well as their interaction effects on leaf area index (LAI), tuber yield, commodity ratio, vitamin C content (VC), crude protein content, water productivity (WP), irrigation water use efficiency (IWUE), internal efficiency of N (IEN), recovery efficiency of N (REN) and net return.

Index	Water (W)	Nitrogen (N)	Year (Y)	W × N	W × Y	N × Y	W × N × Y	Treatment
LAI	*** ^1^	***	NS	NS	NS	NS	NS	***
Tuber yield	***	***	***	NS	***	***	NS	***
Commodity tuber ratio	***	***	***	**	***	***	NS	***
VC	NS	NS	***	NS	NS	NS	NS	NS
Crude protein	***	***	***	NS	NS	NS	NS	***
WP	***	***	**	NS	***	NS	NS	***
IWUE	***	***	***	NS	***	***	NS	NS
IEN	***	***	***	***	***	**	NS	***
REN	***	**	NS	NS	***	NS	NS	***
Net return	***	***	***	NS	***	***	NS	***

^1^ The significant symbols of *** and ** mean significant differences at the level of *p* < 0.001 and *p* < 0.01, respectively, NS means no significant differences.

**Table 2 plants-13-01636-t002:** The application rates of water and nitrogen (N) for different treatments.

Treatment ^1^	Irrigation Amount (mm)		Chemical Nutrients (kg ha^−1^)			Manure Nutrients (kg ha^−1^)		
	2020	2021	N	P_2_O_5_	K_2_O	N	P_2_O_5_	K_2_O
W0N0	0	0	0	100	190	0	0	0
W0N1	0	0	210	100	190	0	0	0
W0N2	0	0	158	90	128	53	10	62
W1N0	230	150	0	100	190	0	0	0
W1N1	230	150	210	100	190	0	0	0
W1N2	230	150	158	90	128	53	10	62
W2N0	286	193	0	100	190	0	0	0
W2N1	286	193	210	100	190	0	0	0
W2N2	286	193	158	90	128	53	10	62

^1^ Water levels: no irrigation (W0), soil-based irrigation (W1), and farmer’s conventional irrigation (W2). Nitrogen levels: no N application (N0), 100% chemical fertilizer N at 210 kg N ha^−1^ application rate (N1), and 25% substitution of chemical N with manure N (N2).

**Table 3 plants-13-01636-t003:** Comprehensive evaluation value and rank of each treatment over two years using the technique for order preference by similarity to ideal solution (TOPSIS) and principal component analysis (PCA) methods.

Treatment ^1^	Comprehensive Evaluation Value		Rank	
	TOPSIS	PCA	TOPSIS	PCA
W0N0	0.197	0.141	9	9
W0N1	0.343	0.313	8	8
W0N2	0.356	0.350	7	7
W1N0	0.591	0.667	5	5
W1N1	0.716	0.719	4	4
W1N2	0.782	0.753	1	1
W2N0	0.512	0.640	6	6
W2N1	0.718	0.741	3	2
W2N2	0.769	0.737	2	3

^1^ Water levels: no irrigation (W0), soil-based irrigation (W1), and farmer’s conventional irrigation (W2). Nitrogen (N) levels: no N application (N0), 100% chemical fertilizer N at 210 kg N ha^−1^ application rate (N1), and 25% substitution of chemical N with manure N (N2).

**Table 4 plants-13-01636-t004:** The lower and upper limits of desirable soil water content (SWC) and soil wetted depth at the different growth stages of the soil-based irrigation regime (W1).

Growth Stage	Lower Limit of SWC	Upper Limit of SWC	Soil Wetted Depth (cm)	Soil Bulk Density (g cm^−3^)
Emergence	60% FC ^1^	100% FC	20	1.34
Tuberization	70% FC	100% FC	30	1.40
Tuber Bulking	70% FC	100% FC	30	1.40
Maturation	60% FC	80% FC	30	1.40

^1^ FC indicates soil field water capacity that is 29.9% in this study.

## Data Availability

Data are contained within the article and the Appendix A; further inquiries can be directed to the corresponding author.

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
