# Peer review of "Synergistic Effects of Soil-Based Irrigation and Manure Substitution for Partial Chemical Fertilizer on Potato Productivity and Profitability in Semiarid Northern China"

_plants, 2024, doi:10.3390/plants13121636_

Round 1
Reviewer 1 Report
Comments and Suggestions for Authors
This study is a two-year experiment in the semiarid region of northern China to investigate the leaf area index (LAI), tuber yield, quality, water and N use efficiency, and net return of potatoes, under the effects of three water irrigation regimes [no irrigation (W0), soil-based irrigation (W1), conventional irrigation (W2)] coupled with three fertilization treatments [no N (N0), 100% chemical N (N1), 25% manure N substitution (N2)].
The trials were good, and many interesting results came out. However, this paper needs further improvements.
The main weaknesses of this paper are as follows:
Abstract: It should briefly summarize the study’s objectives, methods, results, and implications. Consider revisiting the abstract to provide clear conclusions and highlight the practical implications of your findings.
Introduction: Make sure it clearly outlines the study’s objectives, provides context, and introduces the scientific hypothesis. Readers should understand why this research is important and what questions you aim to answer.
Discussion: If you’ve already presented a result in the results section, don’t repeat it in the discussion. Instead, elaborate on its significance.
English: Certain paragraphs are unclear, revise them to enhance clarity. Remember that scientific writing should be accessible to a broad audience.
M&M are presented well. The only problem I noticed in the trial design Fig 1b. It is not a Split plot, it seems to be a Strip plot however the subplot factor should be arranged in vertical strips to the whole plot but it is not. If we consider this design a split plot there is bias because we do not have replications of the main factor. In the case the field is uniform the bias is lower. I suggest deleting the Fig 1b since this design does not exist.

Author Response
Response to Reviewer 1 Comments
The trials were good, and many interesting results came out. However, this paper needs further improvements.
The main weaknesses of this paper are as follows:
Abstract: It should briefly summarize the study’s objectives, methods, results, and implications. Consider revisiting the abstract to provide clear conclusions and highlight the practical implications of your findings.
Thank you very much for your comments. The abstract part has been greatly modified to make the objectives and methods more summarized and the results and conclusions clearer (lines 18-38). Please see the attachment.
Introduction: Make sure it clearly outlines the study’s objectives, provides context, and introduces the scientific hypothesis. Readers should understand why this research is important and what questions you aim to answer.
Thank you for your comments. The introduction has been significantly revised to ensure that the reader has a clearer understanding of the study's background, scientific hypotheses, and objectives.
Discussion: If you’ve already presented a result in the results section, don’t repeat it in the discussion. Instead, elaborate on its significance.
Agreed and revised. In the revised manuscript, we have integrated the results and discussions (lines 87-353) to avoid repeating results and to discuss the results in detail and explain their significance.
English: Certain paragraphs are unclear, revise them to enhance clarity. Remember that scientific writing should be accessible to a broad audience.
Revised. According your comments marked in previous manuscript, the English language have been checked and revised throughout the manuscript.
M&M are presented well. The only problem I noticed in the trial design Fig 1b. It is not a Split plot, it seems to be a Strip plot however the subplot factor should be arranged in vertical strips to the whole plot but it is not. If we consider this design a split plot there is bias because we do not have replications of the main factor. In the case the field is uniform the bias is lower. I suggest deleting the Fig 1b since this design does not exist.
Thanks for pointing out the incorrect description of the trial design. We have revised the description of the experimental arrangement to strip plot (line 373) and deleted Figure 1b.
Please see the attachment for the revised manuscript.

Reviewer 2 Report
Comments and Suggestions for Authors
ID:plants-3011703
Comments:
1. The title is too general. The rsults refer to one experiment. Needs to clarified.
2. Nitrogen is a factor controlled by farmers. Therefore, it cannot be treated as random.
3. The abstract needs to be corrected. A correctly constructed abstract contains: research hypothesis, basic elements of the methodology, basic results and the final conclusion that summarizes the research. The length of the abstract is of 200 words.
4. Keywords cannot coincide with the thematic headings of the article title.
5. Line 42: What about maize?
6. Lines 51-52: This is commonly belevied by farmers, but it is not true. Potato yield and quality depends on the interaction of K and N. K = clay soil.
7. Lines 52-55. Over-interpretation.
8. Lines 80-85: Nothing new. The use of farmyard manure in potato has been known for over 150 years.
9. Lines 100-103: This is part of the analysis of results, not research objectives.
10. Table 1 and 2: The reader is not interested in the research report contained in table 2. This table should be moved to the supplement. The reader is interested in the response of the examined characteristics to the stusied factors, which should operated in the following order:
1) Y × W × N; It concerns two characteristics: ET and N uptake.
2) W × N; It concerns commodity tuber ratio and IEN; In both cases, it is necessary to determine the independence of this interaction on years.
3) The same procedure should be carried out for the remaining interactions of the studied factors.
11. The LAI index depended on W and N, but on interaction between them, including years. So this is how it should be discussed.
12. Lines: 196-295: These are research results, not a discussion.
13. The number of references is too large. reduce by 50%.
General conclusion:
The results section and the discussion section should be be carrierd out in accordance with results of the analysis of variance. For most studied characteritics no W × N or Y × W × N interaction was found. Most characteritics depended on the interaction of Y × W. In some cases, the studied characteristics, for example tuber yield, depended on two interactions: Y × W and Y × N. The question is, which one was dominant? And this needs to be presented in a figure and next discussed.
Round 2
Reviewer 1 Report
Comments and Suggestions for Authors
The authors have made all the corrections suggested in 1st revision.
However, I suggest a change in title from "Soil-based irrigation combined with partial manure substitution for chemical fertilizer improves potato productivity and profitability in semiarid northern China." to
" Soil-based irrigation combined with manure substitution as partial replacement of chemical fertilizer improves potato productivity and profitability in semiarid northern China."
Best Regards
Comments on the Quality of English LanguageModerate editing is needed.
Author Response
However, I suggest a change in title from "Soil-based irrigation combined with partial manure substitution for chemical fertilizer improves potato productivity and profitability in semiarid northern China." to
" Soil-based irrigation combined with manure substitution as partial replacement of chemical fertilizer improves potato productivity and profitability in semiarid northern China."
Thank you for your comments. The title has been revised to “Synergistic effects of soil-based irrigation and manure substitution for partial chemical fertilizer on potato productivity and profitability in semiarid northern China”.
Reviewer 2 Report
Comments and Suggestions for Authors
ID: plants-3011703
Comments:
1. The title in the presented version is actually the conclusion from the study.
2. The abstract needs to be corrected. A correctly constructed abstract contains: research hypothesis, basic elements of the methodology, basic results and the final conclusion that summarizes the research. The length of the abstract is of 200 words.
3. Keywords are descriptions and not thematic headings of the article.
4. Lack of properly formulated research hypothesis.
5. It is advisable to separate the discussion from the Results and Discussion chapter. The article will gain in substantive quality.
6. The conclusion is a summary, not the conclusions of the study.
7. The number of references is too large. 40 references are enough for this article.
